# Link between bilaterality and type 2 inflammation in respiratory epithelial adenomatoid hamartomas: A two-center retrospective cohort study

**Yujin Heo[1], Sang Duk Hong[1], Hyo Yeol Kim[1], Gwanghui Ryu[1], Kyung Won Kwon[2], Yong Gi Jung** [1] *

1 Department of Otorhinolaryngology, Head and Neck Surgery, Samsung Medical Center, Sungkyunkwan University School of Medicine, Seoul, Republic of Korea, 2 Department of Otorhinolaryngology, Head and Neck Surgery, Samsung Changwon Hospital, Sungkyunkwan University School of Medicine, Changwon, Republic of Korea

* ent.jyg@gmail.com

**Data Availability Statement:** All relevant data are within the manuscript and its Supporting Information files.

## Abstract

### Background

Sinonasal respiratory epithelial adenomatoid hamartomas (REAH) are rare benign lesions with diverse clinical characteristics. Their pathogenesis remains unclear, with hypotheses suggesting either an inflammatory process or neoplastic origin. Despite their clinical heterogeneity, a formal subtype classification is lacking. This study analyzes the clinical features of surgically treated sinonasal REAH and investigates disease characteristics based on bilaterality, particularly in relation to type 2 chronic rhinosinusitis.

### Methods

This retrospective analysis included patients with REAH who underwent endoscopic surgery and received pathological confirmation between November 2008 and November 2023. Demographic data, including asthma history, prior sinus surgeries, serum eosinophil levels, and the Japanese Epidemiological Survey of Refractory Eosinophilic Chronic Rhinosinusitis (JESREC) score, were analyzed in relation to REAH bilaterality.

### Results

Among 21 patients (17 males, 4 females; mean age 46.5 years), 11 cases (52.4%) exhibited bilateral origins (REAH_bi) in the superior nasal septum and olfactory cleft, while 10 cases (47.6%) presented as isolated unilateral lesions (REAH_uni). REAH_bi was significantly associated with asthma (p = 0.012), prior sinus surgery (p = 0.002), and inflammatory polyposis (p = 0.002) compared to REAH_uni. Elevated preoperative serum eosinophil levels and JESREC scores were also noted in REAH_bi cases (p = 0.021 and <0.001, respectively). Neither group showed recurrence during a mean follow-up of 11.91 months.

**Funding:** The author(s) received no specific funding for this work.

**Competing interests:** The authors have declared that no competing interests exist.

## Conclusion

Bilaterally originating REAH in the superior nasal septum and olfactory cleft demonstrates pronounced type 2 inflammatory characteristics, suggesting potential differences in pathogenesis compared to unilateral REAH. These findings underscore the need for further investigation into REAH pathophysiology and emphasize the importance of bilaterality in clinical assessment.

## Introduction

Respiratory epithelial adenomatoid hamartomas (REAH) are rare benign neoplasms of the sinonasal tract, pathologically characterized by glandular proliferation of respiratory surface epithelium extending into the submucosal layer [1]. Since Wening and Heffner first described REAH in 1995 [1], reports of this condition have steadily increased over the past two decades. Nevertheless, REAH remains underdiagnosed and is often unfamiliar to many otolaryngologists and pathologists [2–6]. Furthermore, its clinical significance and potential association with type 2 inflammatory diseases, such as asthma, remain insufficiently explored.

The pathogenesis of REAH remains a subject of debate, with two primary hypotheses proposed. One theory suggests that REAH is linked to chronic sinonasal inflammation, a history of asthma, and prior sinus surgeries. Evidence supporting this includes the identification of inflammatory mediators, such as tryptase-producing mast cells, eosinophils, Th9 helper cells, interleukin-9, and eosinophilic cationic proteins, in REAH specimens [7–10]. These findings suggest that REAH may arise through an inflammatory process. Conversely, others hypothesize that REAH may have a neoplastic origin, citing its occasional coexistence with sinonasal malignancies [11]. Additionally, unusually high allelic loss has been observed polypoid sinus tissue [12]. Immunohistochemical analyses of low-grade tubular sinonasal adenocarcinomas associated with REAH have revealed shared characteristics, further supporting a potential neoplastic relationship [13]. Understanding the pathogenesis of REAH is essential for accurate diagnosis and effective management, and its unclear etiology likely contributes to underdiagnosis.

REAH commonly originates from various sites within the sinonasal cavities, with the olfactory cleft and the superior nasal septum being the most frequent locations [5, 14, 15]. The reasons for this predilection remain unclear, though Schertzer *et al.* have discussed a possible relationship with central compartment atopic disease [5]. While existing studies have not extensively analyzed differences in REAH based on origin, clinical observations at our institution suggest that site-specific variations in patient characteristics may exist.

Over the years, our tertiary medical center has treated numerous referred cases of REAH. We have observed that REAH can arise in different sinonasal locations and that clinical characteristics may vary based on their site of origin. These observations prompted us to conduct a comprehensive study examining pathologically confirmed cases of REAH, categorized by their origins, and investigating potential association with clinical features indicative of type 2 chronic rhinosinusitis. By exploring the relationship between REAH origin and clinical presentation, we aim to enhance understanding of this rare and often overlooked neoplasm, ultimately improving diagnosis and patient care.

## Materials and methods

We conducted a retrospective chart review of records between November 1, 2008, and November 1, 2023, involving patients diagnosed with REAH. The pathology database of two tertiary referral hospitals was used to identify cases confirmed through surgical specimens obtained after complete resection. Data collected included sex, age at the time of surgery, previous asthma diagnosis, history of sinonasal surgery, postoperative follow-up period, and recurrence. Asthma diagnosis was based on medical records indicating evaluation by an allergist or pulmonologist and use of inhaled bronchodilators in the absence of other pulmonary conditions.

This study was conducted after obtaining approval from the Institutional Review Board of Samsung Medical Center(No. SMC 2022-06-142). Written informed consent was waived due to the study's retrospective nature using medical records. Data access was restricted to the first and corresponding authors during collection, after which it was anonymized. Data was collected and curated from July 15, 2022, to May 14, 2023, and updated between November 1, 2023, and April 1, 2024. This study is reported as per the Strengthening the Reporting of Observational Studies in Epidemiology (STROBE) guidelines (S1 Appendix).

Preoperative laboratory results, obtained at least three months before surgery, included serum eosinophil percentage and total immunoglobulin E (IgE) levels. Preoperative endoscopic images, intraoperative findings, and medical records were reviewed to assess associations between REAH and sinonasal polyposis. Thin-section (0.6 mm) computed tomography (CT) scans were analyzed by a professional radiologist with over 10 years of experience to determine the location and origin of REAH. Type 2 chronic rhinosinusitis was assessed using the Japanese Epidemiological Survey of Refractory Eosinophilic Chronic Rhinosinusitis (JES-REC) scoring system [16].

Statistical analyses included the student's *t*-test and Mann-Whitney test to compare mean age, preoperative serum eosinophil levels, JESREC score, serum total IgE, and follow-up duration. Fisher's exact test was used to evaluate differences in gender, associated inflammatory polyps, prior surgery, and asthma prevalence. Two-sided p-values <0.05 were considered statistically significant. Analyses were performed using SPSS Statistics for Windows, Version 27.0 (IBM Corporation, Armonk, NY). All data used in the analysis are available in the supplement as S2 Appendix.

## Results

From the pathology datasets of two tertiary referral hospitals, 21 cases of sinonasal REAH were identified. The mean age of patients was 46.5 years (range 14–67 years), with 17 males (81%) and 4 females (19%). Based on preoperative CT findings, cases were categorized into two groups and are shown in Table 1 (Fig 1). Eleven cases (52.4%) with confirmed bilateral origin were classified as REAH_bi; all originating from the superior nasal septum or olfactory cleft. Ten cases (47.6%) with unilateral origin in the nasal cavity or paranasal sinuses, excluding the olfactory cleft or superior nasal septum, were classified as REAH_uni. REAH_bi showed significantly higher rates of prior sinus surgery (90.9% vs. 20.0%, p = 0.002) and asthma (54.5% vs. 0%, p = 0.012) compared to REAH_uni.

Endoscopic findings demonstrated a significantly higher association of inflammatory polyposis in REAH_bi compared to REAH_uni (Fig 2). Pathological review indicated no diagnostic differences between the groups (Fig 3).

REAH_bi cases exhibited significantly higher serum eosinophil levels than REAH_uni (8.32 ± 6.82% vs. 2.60 ± 2.15%, p = 0.021). JESREC scores, used to assess type 2 chronic rhinosinusitis, were also significantly higher in REAH_bi patients(12.91 ± 4.35 vs. 4.75 ± 3.0, p = <0.001). No significant differences were observed in age, sex, symptom duration, serum total

**Table 1. Demographics of enrolled subjects.**

| | REAH_bi (n = 11) | REAH_uni (n = 10) | p-value |
|---|---|---|---|
| Age (years, mean ± SD) | 49.27 ± 11.78 | 43.40 ± 15.41 | 0.336 |
| Gender (Male, n, %) | 8 (72.7) | 9 (90.0) | 0.586 |
| Associated inflammatory polyp (n, %) | 10 (90.9) | 2 (20.0) | **0.002** |
| Symptom duration (>6months, n, %) | 11 (100.0) | 5 (62.5) | 0.058 |
| Previous surgery (n, %) | 10 (90.9) | 2 (20.0) | **0.002** |
| Asthma (n, %) | 6 (54.5) | 0 (0) | **0.012** |
| Serum Eosinophil (%, mean ± SD) | 8.32 ± 6.82 | 2.60 ± 2.15 | **0.021** |
| Serum Total immunoglobulin E | 177.59 ± 158.17 | 81.05 ± 105.90 | 0.231 |
| JESREC score (mean ± SD) | 12.91 ± 4.35 | 4.75 ± 3.01 | <**0.001** |
| Recurrence | - | - | - |
| Follow-up (months, mean±SD) | 16.04 ± 40.53 | 6.23 ± 5.53 | 0.509 |

SD, standard deviation; REAH; Respiratory epithelial adenomatoid hamartoma, REAH_bi; a group of patients with REAH originating from the bilateral superior septum or olfactory cleft, REAH_uni; a group of patients with REAH originating from unilateral sinonasal anatomy apart from the superior septum or olfactory cleft

IgE, or follow-up duration. Inflammatory polyps were pathologically confirmed in 90.9% (10/11) of REAH_bi cases, compared to 20.0% (2/10) of REAH_uni cases (p = 0.002). No recurrences were reported during a mean follow-up period of 11.91 ± 30.81 months.

## Discussion

REAH remain an underdiagnosed disease. Until 2006, only 50 cases were reported, a figure that increased to 600 within a decade [2, 4–6]. Issa *et al*. demonstrated that 47.4% of 114 patients with olfactory cleft enlargement, initially treated for chronic rhinosinusitis with nasal polyps, were later histologically confirmed to have REAH [3]. This underdiagnosis is largely attributed to the lack of efficient diagnostic methods before surgical confirmation. Most

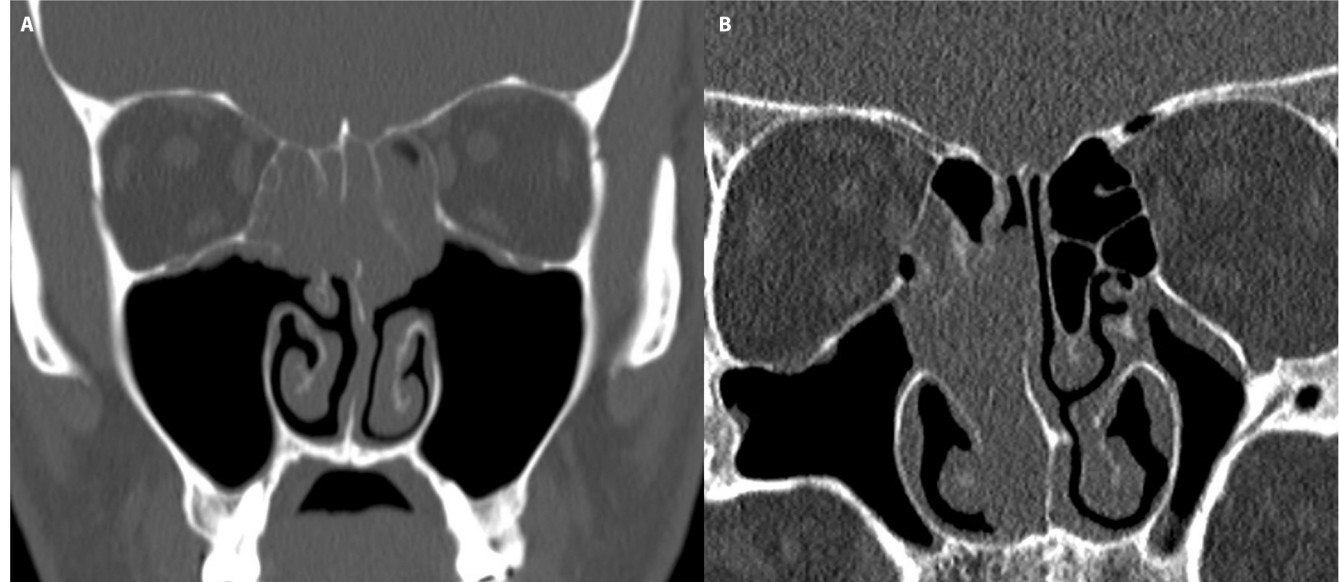

**Fig 1. Preoperative coronal computed tomography scans.** REAH_bi (A) reveals a hypodense expansile lesion involving both sides of the olfactory cleft, extending to the turbinates. REAH_uni (B) shows a mass in the right nasal cavity, extending to the septum sparing the olfactory cleft.

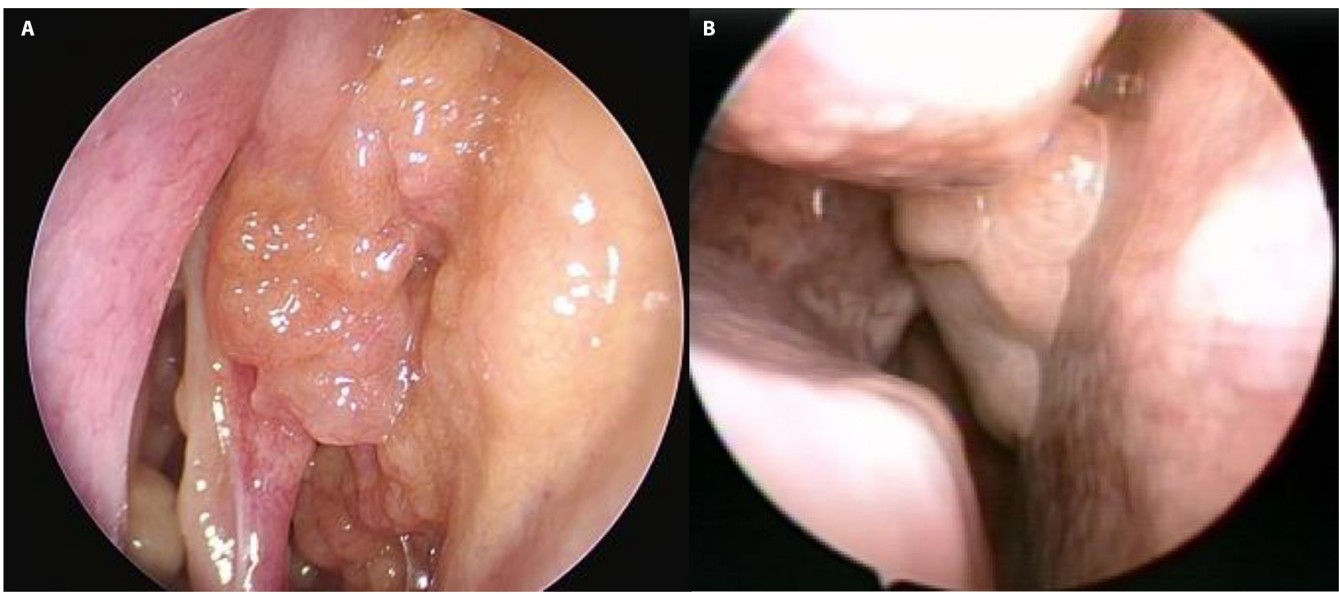

**Fig 2. Nasal endoscopic images of respiratory epithelial adenomatoid hamartomas.** REAH_bi (A) demonstrates indistinct boundaries between the lesion and inflammatory polyposis, whereas REAH_uni (B) shows a smooth mucosa-covered polypoid mass on the posterior nasal septum without other nasal lesions.

patients present with nonspecific obstruction-related symptoms-such as nasal obstruction, discharge, olfactory impairment, or facial pressure-clinically mimicking chronic rhinosinusitis [15]. Conversely, REAH may sometimes be mistaken for tumorous condition, such as inverted papilloma, leading to unnecessary radical surgery [17–19]. Additionally, the frequent association of REAH with nasal polyposis further complicates pre-surgical diagnosis [20–22]. Endoscopic examinations often reveal nasal masses of varying size and color, making gross identification of REAH challenging [23].

To better classify and understand this disease, we hypothesized that REAH originating bilaterally from the superior septum and olfactory cleft (REAH_bi) exhibits distinct characteristics compared to unilateral lesions sparing the olfactory cleft (REAH_uni). Evidence supporting this includes a significantly higher prevalence of prior sinus surgery and asthma in the REAH_bi group, alongside elevated preoperative serum eosinophil level and JESREC score-markers of type 2 inflammation. These findings suggest that REAH_bi is more likely to have a reactive inflammatory etiology compared to REAH_uni.

Previous studies have explored diagnostic and classification approaches for REAH. REAH is often classified based on concurrent sinonasal inflammatory disorders, such as inflammatory polyposis [9, 15, 23], which are associated with more severe symptoms and a heavier disease burden [14]. Other research has emphasized the diagnostic importance of lesion location, primarily involving the posterior nasal septum and olfactory cleft [5, 9]. Hawley *et al.* demonstrated that an olfactory cleft width >10mm on coronal CT scans yielded 88% sensitivity for detecting REAH [9]. However, this method has limitations as REAH may also occur in other sinonasal sites and the nasopharynx [24–27]. To our knowledge, differences in pathogenesis based on REAH origin have not been previously investigated. Moreover, current clinical management does not consider lesion location or concurrent inflammatory conditions. Differences in patient characteristics based on lesion origin may offer crucial insights into REAH pathogenesis and serve as a foundation for improving diagnostic and therapeutic strategies.

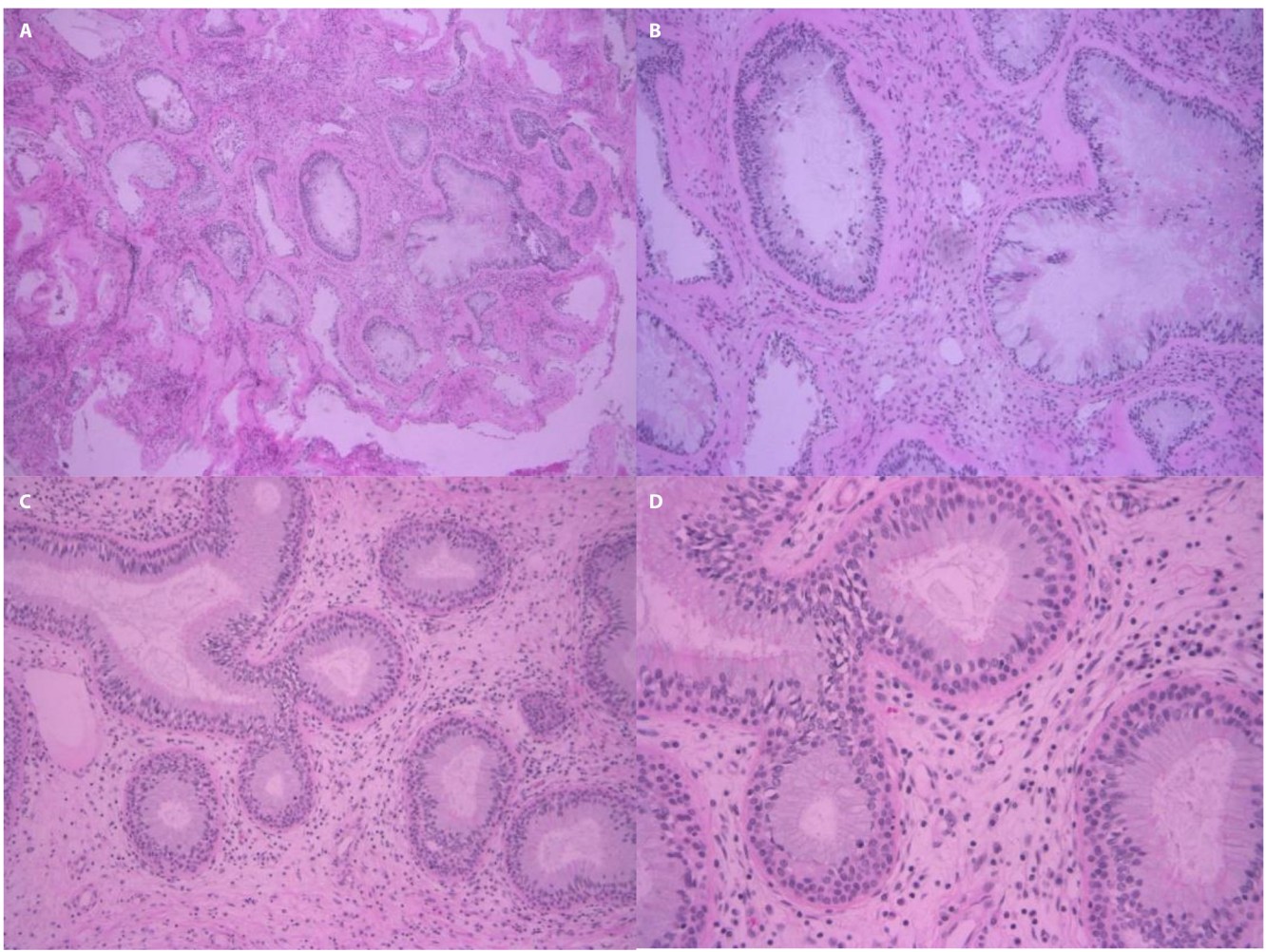

**Fig 3. Microscopic findings of respiratory epithelial adenomatoid hamartomas.** Both REAH_bi (A, B) and REAH_uni (C, D) display similar histological features, including glandular proliferation of pseudostratified ciliated respiratory epithelium and edematous stroma with scattered inflammatory cells (Hematoxylin and Eosin, magnification A: x100, B: x200, C: x100, D: x200).

## Limitations

This study has several limitations, primarily due to its retrospective design and small sample size of 21 patients. The small number of cases precluded formal sample size calculations and may limit the statistical power of our findings. However, it is essential to note that REAH remains an underdiagnosed and rare entity in clinical practice, making large cohort studies challenging. Despite these limitations, our series of 21 cases represents a meaningful contribution to the current literature on REAH, particularly regarding the characterization of bilateral versus unilateral disease patterns. Future prospective studies, including a larger cohort based on the origin of REAH, would help to further validate our findings, investigate the pathogenesis of REAH, and explore effective diagnostic methods and treatments based on its entity.

## Conclusion

The REAH_bi group demonstrated more frequent typical type 2 inflammatory characteristics compared to the REAH_uni group. While histological differences were absent between the

two types, their distinct clinical characteristics-such as a history of asthma and the coexistence of inflammatory polyps in the REAH_bi group- highlight potential differences in underlying pathophysiology. These findings suggest that REAH may not present a uniform disease entity but instead comprise lesions with distinct inflammatory or neoplastic features based on their origins.

This study provides valuable insights into the classification and characterization of REAH according to its site of origin. Further research with larger cohorts is essential to validate these findings and to develop more effective diagnostic and therapeutic approaches for managing REAH.

## Supporting information

**S1 Appendix. STROBE statement—checklist of items that should be included in reports of cohort studies.**
(DOCX)

**S2 Appendix. Data set of patents diagnosed with respiratory epithelial adenomatoid hamartomas.**
(DOCX)

## Author Contributions

**Conceptualization:** Yong Gi Jung.

**Data curation:** Yujin Heo, Sang Duk Hong, Hyo Yeol Kim, Gwanghui Ryu, Kyung Won Kwon, Yong Gi Jung.

**Formal analysis:** Yujin Heo, Yong Gi Jung.

**Methodology:** Yujin Heo, Yong Gi Jung.

**Visualization:** Yong Gi Jung.

**Writing – original draft:** Yujin Heo, Sang Duk Hong, Hyo Yeol Kim, Gwanghui Ryu, Kyung Won Kwon, Yong Gi Jung.

**Writing – review & editing:** Yujin Heo, Yong Gi Jung.

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
