## [Decision Letter · Decision Letter 0]

18 Nov 2024

PONE-D-24-26646Link Between Bilaterality and Type 2 Inflammation in Respiratory Epithelial Adenomatoid Hamartomas: A Two-center Retrospective Cohort StudyPLOS ONE

Dear Dr. Jung,

Thank you for submitting your manuscript to PLOS ONE. After careful consideration, we feel that it has merit but does not fully meet PLOS ONE’s publication criteria as it currently stands. Therefore, we invite you to submit a revised version of the manuscript that addresses the points raised during the review process.

This manuscript must be consistently edited either for English and mistakes contained in it.In addition the number of cases is considerably low and this must be written in the Limitations at the end of the Discussion in a separate subheading.

We look forward to receiving your revised manuscript.

Kind regards,

Gianpaolo Papaccio, M.D., Ph.D.

Academic Editor

PLOS ONE

Journal Requirements:

Additional Editor Comments :

This study is based on a small sample size that greatly limits the study.

This must be clearly written in the Limitations of the study as a subheading before the Conclusions.

In addition, the manuscript is plenty of mistakes either of language and content, as highlighted by reviewer #2.

The Table if not substituted by a Figure must change position.

The Discussion must not contain Results.

The English requires a deep revision

Reviewers' comments:

Reviewer's Responses to Questions

**Comments to the Author**

1. Is the manuscript technically sound, and do the data support the conclusions?

Reviewer #1: Yes

Reviewer #2: Yes

2. Has the statistical analysis been performed appropriately and rigorously? 

Reviewer #1: Yes

Reviewer #2: Yes

3. Have the authors made all data underlying the findings in their manuscript fully available?

Reviewer #1: Yes

Reviewer #2: Yes

4. Is the manuscript presented in an intelligible fashion and written in standard English?

Reviewer #1: Yes

Reviewer #2: Yes

5. Review Comments to the Author

Reviewer #1: In the study, assuming that REAH can be located in different places in the sinonasal anatomy and that its clinical features may vary depending on the region of origin, a comprehensive study examining the pathologically confirmed clinical REAH series was conducted to investigate their potential relationship with clinical features indicative of type 2 chronic rhinosinusitis.

I suggest the authors to explain the calculation for sample size (G-power analysis) in the Methods section.

Reviewer #2: This retrospective study discussed the findings of respiratory epithelial adenomatoid hamartoma (REAH) and found that bilateral REAH exhibited type 2 inflammatory features more commonly compared to unilateral REAH. Since REAH is a rare disease often masked by other clinical features, such as nasal polyps, findings from this study provide a new perspective that can aid in the diagnosis of REAH. Also mentioned in the manuscript, the small sample size is one of the limitations of the study. Overall, the manuscript is well-written and easy to understand.

There are a few comments as below:

1. Abstract: (line 41) “…we report our treatment experience…”. The routine treatment for REAH is surgery. In the manuscript, treatment is not the main focus; the results and discussion are about the patients’ clinical history and characteristics. Suggest to add details of the treatment given or to edit the sentence.

2. Materials and methods: for the date format, suggest spelling the month. For example, 1 November 2008 or November 1, 2008.

3. Results: (page 8) Table 1 seems to be located in the middle of the text, not at the end of the paragraph. Suggest to check the format and text alignment to avoid confusion.

4. Discussion: (line 193 – 209). These are the results of endoscopic examination and microscopic appearance, which are not covered in the results section. Suggest to move it to the results.

5. References:

i. (line 83, introduction) Reference no.13 described the details of immunohistochemistry findings of sinonasal adenocarcinomas, which are associated with REAH. However, the study did not include a genotypic or allelic study, hence not suitable to be referred for allelic loss.

ii. (line 173, discussion) Reference no.3: In the article by Issa et al, 47.4% of the patients with polyposis were later confirmed to have REAH. Suggest to revise the number or sentence.

6. PLOS authors have the option to publish the peer review history of their article (what does this mean?). If published, this will include your full peer review and any attached files.

Reviewer #1: No

Reviewer #2: No

---

## [Author Response · Author response to Decision Letter 0]

14 Jan 2025

Submission of Revised Manuscript PONE-D-24-26646

Dear Editor and Reviewers,

I am writing to submit the revised version of our manuscript titled “Link between bilaterality and type 2 inflammation in Respiratory Epithelial Adenomatoid Hamartomas: A two-center retrospective cohort study” (PONE-D-24-26646).

We sincerely thank you and the reviewers for the constructive and detailed feedback provided on our manuscript. These valuable comments have allowed us to improve our work's clarity, depth, and overall quality.

In the revised manuscript, we have addressed all the points raised by the reviewers and the editorial team, incorporating their suggestions to enhance the study’s scientific rigor. The key changes include:

1. Revision of the manuscript’s English language for clarity and grammar, as recommended by the editorial team. We have originally gone through a professional English editing process provided by E World Editing and will attach the certificate as part of this submission.

2. Addition of a "Limitations" subheading in the Discussion section to explicitly discuss the small sample size and its impact on the study’s findings.

3. Editing the manuscript to ensure compliance with PLOS ONE’s style requirements, including proper naming and adherence to formatting templates.

4. Inclusion of a full ethics statement in the Methods section, specifying the Institutional Review Board (IRB) approval (No. SMC 2022-06-142) and the waiver of informed consent due to the retrospective nature of the study.

5. Uploading the raw data set as a supplementary file to meet the journal’s Data Availability Statement requirements. This data includes all values behind the reported means, standard deviations, and other measures.

6. Ensuring that all figure files have been uploaded and verified using the Preflight Analysis and Conversion Engine (PACE) digital diagnostic tool, as required by the journal.

7. Relocation of Table 1 to align with the corresponding text for improved readability.

8. Reorganization of endoscopic and microscopic findings from the Discussion section to the Results section for proper manuscript structure.

To facilitate your review, we have highlighted all the changes in the revised manuscript and provided a detailed, point-by-point response to the reviewers’ comments.

We sincerely appreciate your understanding and flexibility in extending this revision. The ongoing medical crisis in South Korea and an unusually high clinical workload posed significant challenges, but your patience allowed us to complete the revisions thoroughly and carefully.

Please find the following documents attached for your consideration:

1. The revised manuscript with highlighted changes.

2. A comprehensive response to the reviewers’ comments.

3. The English editing certificate issued by E World Editing.

4. The raw data set uploaded as a supplementary file.

We hope that the revisions satisfactorily address all concerns and align with PLOS ONE's expectations. Should further clarification or additional modifications be necessary, we are more than willing to address them promptly.

Once again, we extend our heartfelt gratitude for your support and the opportunity to contribute to the journal. We look forward to your feedback on our revised manuscript.

Kind regards,

Yong Gi Jung

POINT-BY-POINT RESPONSE

EDITOR

This manuscript must be consistently edited either for English and mistakes contained in it.

ANSWER: 

We appreciate your generous comments on improving this manuscript. We revised the manuscript to ensure proper English editing and corrected the identified mistakes. The parts edited with proper English are highlighted in blue.

EDITOR

In addition, the number of cases is considerably low and this must be written in the Limitations at the end of the Discussion in a separate subheading.

ANSWER: 

We have updated a "Limitations" subheading at the end of the Discussion section explicitly stating the impact of the small sample size. Thank you for your insightful comment.

CORRECTION: 

Discussion (Page 11, Line 211-221)

Limitations

This study has several limitations, primarily due to its retrospective design and small sample size of 21 patients. The small number of cases precluded formal sample size calculations and may limit the statistical power of our findings. However, it is essential to note that REAH remains an underdiagnosed and rare entity in clinical practice, making large cohort studies challenging. Despite these limitations, our series of 21 cases represents a meaningful contribution to the current literature on REAH, particularly regarding the characterization of bilateral versus unilateral disease patterns. Future prospective studies, including a larger cohort based on the origin of REAH, would help to further validate our findings, investigate the pathogenesis of REAH, and explore effective diagnostic methods and treatments based on its entity.

EDITOR

This study is based on a small sample size that greatly limits the study.

This must be clearly written in the Limitations of the study as a subheading before the Conclusions.

ANSWER: 

Thank you for raising this important point. We have updated a "Limitations" subheading at the end of the Discussion section explicitly stating the impact of the small sample size. 

CORRECTION: 

Discussion (Page 11, Line 211-221)

Limitations

This study has several limitations, primarily due to its retrospective design and small sample size of 21 patients. The small number of cases precluded formal sample size calculations and may limit the statistical power of our findings. However, it is essential to note that REAH remains an underdiagnosed and rare entity in clinical practice, making large cohort studies challenging. Despite these limitations, our series of 21 cases represents a meaningful contribution to the current literature on REAH, particularly regarding the characterization of bilateral versus unilateral disease patterns. Future prospective studies, including a larger cohort based on the origin of REAH, would help to further validate our findings, investigate the pathogenesis of REAH, and explore effective diagnostic methods and treatments based on its entity.

EDITOR

In addition, the manuscript is plenty of mistakes either of language and content, as highlighted by reviewer #2.

ANSWER: 

We appreciate your generous comments to improve this manuscript. We thoroughly revised the manuscript to correct all language and content issues highlighted by Reviewer #2.

EDITOR

The Table if not substituted by a Figure must change position.

ANSWER: 

Thank you for your thoughtful comment. We repositioned Table 1 appropriately within the text for clarity and ensured it aligned with the surrounding discussion.

CORRECTION: 

Results (Page 8, line 142-146)

Table 1. Demographics of enrolled subjects

EDITOR

The Discussion must not contain Results.

ANSWER: 

We appreciate your valuable feedback. We moved all result-related content from the Discussion to the Results section to adhere to the correct manuscript structure.

CORRECTION: 

Results (Page 8, line 153-166)

Endoscopic findings demonstrated a significantly higher association of inflammatory polyposis in REAH_bi compared to REAH_uni (Fig 2). Pathological review indicated no diagnostic differences between the groups (Fig 3). 

Figure 2. Nasal endoscopic images of Respiratory Epithelial Adenomatoid Hamartomas

REAH_bi (A) demonstrates indistinct boundaries between the lesion and inflammatory polyposis, whereas REAH_uni (B) shows a smooth mucosa-covered polypoid mass on the posterior nasal septum without other nasal lesions.

Figure 3. Microscopic findings of Respiratory Epithelial Adenomatoid Hamartomas

Both REAH_bi (A, B) and REAH_uni (C, D) display similar histological features, including glandular proliferation of pseudostratified ciliated respiratory epithelium and edematous stroma with scattered inflammatory cells (Hematoxylin and Eosin, magnification A: x100, B: x200, C: x100, D: x200).

EDITOR

The English requires a deep revision

ANSWER: 

We appreciate your comment. We conducted a thorough English revision of the manuscript using a professional editing service. The revised parts are highlighted in blue.

REVIEWER #1

In the study, assuming that REAH can be located in different places in the sinonasal anatomy and that its clinical features may vary depending on the region of origin, a comprehensive study examining the pathologically confirmed clinical REAH series was conducted to investigate their potential relationship with clinical features indicative of type 2 chronic rhinosinusitis.

I suggest the authors to explain the calculation for sample size (G-power analysis) in the Methods section.

ANSWER: 

Thank you for your insightful feedback and for the opportunity to further clarify our study methodology. Regarding your request for a G-Power analysis for sample size calculation, we would like to explain why such an analysis was not feasible in this context.

Respiratory epithelial adenomatoid hamartoma (REAH) is an underdiagnosed pathological entity, with limited cases reported in the literature and no comprehensive systematic reviews available to provide reliable estimates for effect sizes or prevalence.(reference) Our study is based on a retrospective review of all available cases treated and pathologically confirmed at our center, one of the largest tertiary referral hospitals in South Korea, over a 15-year period (November 2008 to November 2023), comprising 21 patients. Despite our institution's status as a major tertiary referral center, this limited number of cases reflects both the rarity of REAH and potential underdiagnosis due to limited awareness among pathologists. This constitutes the entire sample of this rare disease entity from our institution. Despite our efforts to include as many cases as possible, we kindly ask reviewers to understand the limited sample size in this study, which reflects the inherent rarity of this condition.

Given the rarity of REAH and the absence of prior systematic data, there are several challenges in applying G-Power analysis for this study:

1. No Established Effect Size

- G-Power calculations require an estimated effect size, typically derived from prior studies or systematic reviews. (ref.) In the absence of such data for REAH, any assumed effect size would be speculative and potentially misleading.

2. Retrospective Nature

- Our study reflects a complete retrospective cohort of patients treated within a specific timeframe, making it impractical to predefine or adjust the sample size based on power calculations.

3. Rarity of Disease

- REAH is an underdiagnosed disease entity, which inherently limits sample sizes, even at specialized centers. As such, the 21 cases represent not a subset but the full population available for study, making external sample size calculations inapplicable.

To address this, we have explicitly emphasized the scarcity of our sample size and the unavailability of performing G-Power analysis in the Limitations section of the Discussion. This addition acknowledges the inherent constraints of studying a rare disease and reinforces the need for future multicenter studies or systematic reviews to validate and expand upon our findings. We sincerely appreciate your valuable feedback to enhance the quality of our manuscript.

(Reference)

Davison WL, Pearlman AN, Donatelli LA, Conley LM. Respiratory epithelial adenomatoid hamartomas: An increasingly common diagnosis in the setting of nasal polyps. Am J Rhinol Allergy. 2016; 30: 139-146. https://doi.org/10.2500/ajra.2016.30.4338 PMID: 27456590

Issa MJA, Oliveira VRR, Nunes FB, Vasconcelos LOG, Souza LFB, Cherobin GB, et al. Prevalence of respiratory epithelial adenomatoid hamartomas (REAH) associated with nasal polyposis: an epidemiological study - how to diagnose. Braz J Otorhinolaryngol. 2022; 88 Suppl 5: S57-S62. https://doi.org/10.1016/j.bjorl.2021.09.009 PMID: 34844870

Lima NB, Jankowski R, Georgel T, Grignon B, Guillemin F, Vignaud JM. Respiratory adenomatoid hamartoma must be suspected on CT-scan enlargement of the olfactory clefts. Rhinology. 2006; 44: 264-269. PMID: 17216743

CORRECTION: 

Discussion (Page 11, Line 211-221)

Limitations

This study has several limitations, primarily due to its retrospective design and small sample size of 21 patients. The small number of cases precluded formal sample size calculations and may limit the statistical power of our findings. However, it is important to note that REAH remains an underdiagnosed and rare entity in clinical practice, making large cohort studies challenging. Despite these limitations, our series of 21 cases represents a meaningful contribution to the current literature on REAH, particularly regarding the characterization of bilateral versus unilateral disease patterns. Future prospective studies, including a larger cohort based on the origin of REAH, would help to further validate our findings, investigate the pathogenesis of REAH, and explore effective diagnostic methods and treatments based on its entity.

REVIEWER #2

Comments and Suggestions for Authors

This retrospective study discussed the findings of respiratory epithelial adenomatoid hamartoma (REAH) and found that bilateral REAH exhibited type 2 inflammatory features more commonly compared to unilateral REAH. Since REAH is a rare disease often masked by other clinical features, such as nasal polyps, findings from this study provide a new perspective that can aid in the diagnosis of REAH. Also mentioned in the manuscript, the small sample size is one of the limitations of the study. Overall, the manuscript is well-written and easy to understand.

There are a few comments as below:

1. Abstract: (line 41) “…we report our treatment experience…”. The routine treatment for REAH is surgery. In the manuscript, treatment is not the main focus; the results and discussion are about the patients’ clinical history and characteristics. Suggest to add details of the treatment given or to edit the sentence.

ANSWER: 

We are grateful for your insightful comments and recommendation to refine our work. We revised the abstract to better reflect the study focus and avoid overemphasis on treatment. The updated sentence emphasizes the clinical experience and disease characteristics.

CORRECTION: 

Abstract (Page 3, line 38-40) 

This study analyzes the clinical features of surgically treated sinonasal REAH and investigates disease characteristics based on bilaterality, particularly in relation to type 2 chronic rhinosinusitis.

REVIEWER #2

2. Materials and methods: for the date format, suggest spelling the month. For example, 1 November 2008 or November 1, 2008.

ANSWER: 

We appreciate your valuable feedback. We adjusted the date format throughout the manuscript to spell out the month consistently.

CORRECTION: 

Materials and Methods (Page 6, line 102-105) 

We conducted a retrospective chart review of records between November 1, 2008, and November 1, 2023, involving patients diagnosed with REAH.

Materials and Methods (Page 

---

## [Editor Report · Decision Letter 1]

16 Jan 2025

Link between bilaterality and type 2 inflammation in Respiratory Epithelial Adenomatoid Hamartomas: A two-center retrospective cohort study

PONE-D-24-26646R1

Dear Dr. Jung,

We’re pleased to inform you that your manuscript has been judged scientifically suitable for publication and will be formally accepted for publication once it meets all outstanding technical requirements.

Kind regards,

Gianpaolo Papaccio, M.D., Ph.D.

Academic Editor

PLOS ONE

Additional Editor Comments (optional):

All the comments have been accepted.

During the proof editing proceed the scientific english will be checked and ameliorated.
---

## [Editor Report · Acceptance letter]

24 Jan 2025

PONE-D-24-26646R1 

PLOS ONE

Dear Dr. Jung, 

I'm pleased to inform you that your manuscript has been deemed suitable for publication in PLOS ONE. Congratulations! Your manuscript is now being handed over to our production team.

Kind regards, 

on behalf of

Prof. Gianpaolo Papaccio 

Academic Editor

PLOS ONE